# Prevalence of Dental Anomalies in Taiwanese Children with Cleft Lip and Cleft Palate

**DOI:** 10.3390/jpm12101708

**Published:** 2022-10-13

**Authors:** Chin-Han Chang, Chi-Hua Chang, Jui-Pin Lai, Shiu-Shiung Lin, Yu-Jen Chang

**Affiliations:** 1Department of Craniofacial Orthodontics, Department of Dentistry, Kaohsiung Chang Gung Memorial Hospital and Chang Gung University College of Medicine, 123 Dapi Rd. Niaosong District, Kaohsiung 833401, Taiwan; 2Department of Oral Surgery, Kaohsiung Chang Gung Memorial Hospital and Chang Gung University College of Medicine, 123 Dapi Rd. Niaosong District, Kaohsiung 833401, Taiwan; 3Department of Plastic and Reconstructive Surgery, Kaohsiung Chang Gung Memorial Hospital and Chang Gung University College of Medicine, 123 Dapi Rd. Niaosong District, Kaohsiung 833401, Taiwan

**Keywords:** cleft lip and palate, tooth anomalies, tooth agenesis, alveolar bone graft

## Abstract

The aim of this study was to investigate the prevalence of tooth agenesis, tooth malformation, and eruption patterns of upper canines/first premolars in Taiwanese children. A total of 132 cleft lip and cleft palate (CLCP) patients (82 boys and 50 girls) underwent alveolar bone grafting (ABG) between 2012 and 2022. The patients’ dental records and X-ray images were inspected. We examined dental anomalies, including congenital missing teeth, microdontia, and transposition from the upper canines to the upper first premolars in these CLCP patients. Additionally, we investigated the mean ABG operation age (9.27 ± 0.76 years) of our patient; 40.9% of them received pre-ABG orthodontic treatment at 8.72 ± 0.70 years. Among the 132 cleft subjects, the prevalence of tooth agenesis is 73.5% (97/132). The most frequently missing teeth are the maxillary lateral incisors (right side: 46.2%; left side: 47.0%). In this study, microdontia are found in all the upper incisors, of which the highest percentage (18.9%) is observed in the upper left lateral incisors. The prevalence of upper canine and first premolar transposition is 10.6%. The pattern of tooth agenesis and microdontia of the upper lateral incisors shows a strong correlation with the cleft sites of these CLCP patients in our study. These results may support the idea that the patterns of dental anomalies in CLCP patients are region-specific.

## 1. Introduction

Among all congenital craniofacial deformities, cleft lip and cleft palate (CLCP) are the most commonly prevalent worldwide [1]. Patients may present with varying degrees of influence on speech, hearing, masticatory function, skeletal and dentoalveolar configuration, and social developmental disorders, depending on the severity of the defect.

The causes of oral cleft are multifactorial and can be categorized into genetic and environmental factors. Research shows that Msx1 and PAX 9 are the two main genes that affect tooth and alveolar bone development, resulting in oral clefts and even tooth agenesis [2,3,4,5,6,7,8]. Environmental factors, such as medications, malnutrition, toxins, and hormonal disorders are reported [9]. Interactions between these factors lead to the failure of fusion of the medial nasal process and maxillary process in the 6th week of intrauterine life.

Dental anomalies, including congenital missing teeth, microdontia, delayed eruption, supernumerary teeth, supplemental teeth, and transposition of the maxillary canines and first premolars, are frequently observed in patients with craniofacial cleft. Congenital missing teeth is a common abnormality defined as the developmental absence of teeth excluding third molars. The frequency of patients with congenitally missing teeth is 1.6–9.6%, as reported by Graber [10]. Microdontia is a condition in which one or more teeth appear dimensionally smaller than the usual limits of variation [11]. When a tooth emerges into the oral cavity at a time that is significantly deviated from the normal eruption time established for different sexes, races, and ethnicities, it is diagnosed as a tooth with delayed eruption [12]. Tooth transposition is manifested by an interchange in the position of two permanent adjacent teeth in the same quadrant of the dental arch. When the affected teeth are completely transposed, complete transposition is observed. If the roots remain in a normal position with only transposed crowns, incomplete transposition is defined [13]. Tooth agenesis is most commonly observed in patients with CLCP [14,15]. Maxillary lateral incisors have the highest missing prevalence, ranging from 56.1% to 74% in the cleft area [16]. Additionally, dental anomalies are more commonly found in the cleft area than in the non-cleft area, and affect the permanent dentition more than the primary dentition [17,18,19,20].

The malformation of teeth tends to result in discrepancy of tooth alignment, hence, CLCP patients require early interdisciplinary rehabilitations to normalize the morphology and masticatory function. The aim of this study was to investigate the prevalence of tooth agenesis, malformation of teeth, and the eruption pattern of upper canines/first premolars in Taiwanese children with CLCP at the time they underwent an alveolar bone grafting (ABG) operation to optimize the position of dentoalveolar structure.

## 2. Materials and Methods

We selected patients with cleft lip and palate enrolled at the Craniofacial Center, Kaohsiung Chang Gung Memorial Hospital, Taiwan, between 2012 and 2022.

The inclusion criteria were: (1) non-syndromic cleft lip and palate (including unilateral and bilateral) in Taiwanese children who received ABG operation; (2) no extraction of permanent dentition prior to the initial screening; and (3) the collection of hospital records, intraoral pictures, study models, and X-ray imaging were available.

The exclusion criteria were as follows: (1) severe craniofacial syndromic cleft lip and palate (e.g., hemifacial microsomia, Pierre Robin syndrome, and Van der Woude syndrome); and (2) history of permanent tooth extraction.

The types of CLCP included both complete and incomplete primary cleft palates, if the alveolar cleft or the thinning alveolus with localized gingival cleft was present. Dental and operative records, intraoral/extraoral pictures, study models, and X-ray imaging, including panoramic radiographs, periapical films of anterior teeth, and upper occlusal films, were constantly and separately inspected by two authors (Chin-Han Chang and Yu-Jen Chang) in two different occasions with an interval of 3 months. The sequence of data inspection was randomized for each occasion. The observed data were then cross-evaluated to ensure that the results of the investigation regarding dental abnormalities were accurately assessed. No permanent dentition extraction was performed before the initial screening. Patients who required orthodontic treatment of the upper anterior teeth before ABG were treated by the Department of Orthodontics, Kaohsiung Chang Gung Memorial Hospital, Taiwan.

We investigated and examined several dental anomalies, including congenital missing teeth, microdontia, and transposition of the upper canines to the upper first premolars, in patients with CLCP. We recorded the average ABG operation age of these patients and investigated the percentage of patients who received pre-ABG orthodontic treatment.

This study was carried out in accordance with the Code of Ethics of the World Medical Association (Declaration of Helsinki) and was approved by the Institutional Review Board of the Chang Gung Medical Foundation (IRB No: 202201360B0).

## 3. Statistical Analysis

Dental anomalies were inspected on panoramic films and study models properly via intra- and inter-observer agreement on two different occasions. Therefore, misinterpretation and misidentification can truly be eliminated, thereby increasing the accuracy of the actual abnormality patterns of dentition. The number of each tooth site and percentage were counted to characterize the dental anomalies.

Statistical analysis of the data was performed using the SPSS software (version 20.0; SPSS Inc., Chicago, IL, USA). Chi-square test was used to analyze the relationship between gender and cleft sides, and the level of significance was set at *p*-value < 0.05.

## 4. Results

In this study, 135 patients with CLCP were initially investigated, of which three syndromic patients (hemifacial microsomia, Pierre Robin syndrome, and Van der Woude syndrome) were excluded. Therefore, a total of 132 patients with cleft lip and palate underwent alveolar bone grafting; 82 were boys (62.1%) and 50 were girls (37.8%). The mean age when receiving ABG was 9.27 ± 0.76 y with a range from 7.04 y to 12.58 y (Table 1). Regarding the distribution of cleft sites, 103 of the 132 (78%) patients present with unilateral; 47 patients (35.6%) present with a cleft on the right side, while 56 patients present with a cleft (42.4%) on the left side. A total of 29 of the 132 (22%) patients present with bilateral cleft lip and palate (Table 2). Despite the higher tendency of male dominance noted in unilateral CLCP and bilateral CLCP patients, the chi-square test shows no significant relationship between sex and the number of affected sides (*p* > 0.05) (Table 3). In addition, no significant difference is found in the chi-square test between the sex and the affected side (*p* > 0.05) (Table 4). Among these 132 patients, 54 (40.9%) received pre-ABG orthodontic treatment at an average age of 8.72 ± 0.70 and underwent ABG at a mean age of 9.13 ± 0.70 y (Table 5).

Among the 132 CLCP, the prevalence of tooth agenesis is 73.5% (97/132). The most frequently missing teeth are the maxillary lateral incisors (right side: 46.2%, left side: 47.0%). One patient (0.8%) is found to present with oligodontia, which is defined as agenesis of six or more teeth. Figure 1 shows the distribution and frequency of missing teeth in all four quadrants of the arch form. Maxillary lateral incisors are the most frequently missing teeth (right side: 46.2%, left side: 47.0%), which corresponds to the cleft site (unilateral or bilateral), followed by maxillary second premolars and mandibular right lateral incisors. In addition, missing maxillary and mandibular central incisors, maxillary canines, first premolars, and second molars are also noted in the agenesis pattern.

The prevalence of microdontia among 132 patients is 28%. In this study, microdontia are found in all the upper incisors. Among these, the upper left lateral incisors have the highest percentage (18.9%), while the upper right lateral incisors are 11.4% (Figure 2). Most cases of microdontia coincide with the corresponding cleft site.

For upper canine and first premolar transposition, the prevalence is 14 subjects (10.6%) among the 132 CLCP patients. Seven patients (5.3%) present with transposition ipsilateral to the cleft site and six patients (4.5%) present with transposition contralateral to the cleft site. Only one patient (0.8%) presents with bilateral transposition (Table 6).

## 5. Discussion

Previous studies report differences in the incidence of cleft lip and palate among races. Ranging from 1 to 7 in every 1000 newborns, the prevalence of clefts in Asians is much higher than in other races [21]. According to Wei and Chen, the prevalence of CLCP in Taiwan is approximately 0.192% [22], while other studies report a prevalence of 0.06% in the United States [23] and 0.18% in both Korea and China [24,25].

Cleft lip and cleft lip with palate are more common in men. The cleft palate is found more frequently in women. Kim et al. [24] report the ratio of cleft lip and palate in men to women as 2.5:1, and Cooper et al. [26] report it as 1.6:1. In our study, the ratio of men to women patients receiving ABG is 1.64:1 (Table 1), which also shows a higher tendency for male dominance.

Unilateral CLCP is more common than bilateral CLCP. Yilmaz et al. [27] report the prevalence of bilateral cleft lip and palate to be 25.5%. Our study shows similar results for bilateral cleft sites, 22% of whom receive 22% ABG. For patients with unilateral cleft, involvement of the left side is more common than that of the right side. Two studies (Wilson [28] and Drillien et al. [29]) show a unilateral cleft prevalence of 60% on the left side. Fraser [30] reports 66.6% of left-sided clefts in unilateral cleft patients. Our results show that the prevalence of left-sided clefts is 54.4% in patients with unilateral cleft with ABG and 42.4% of all cleft types (Table 2).

The timing of ABG at the mixed dentition stage is widely accepted. At approximately 9 y of age, ABG allows bone growth to complete with minimal maxillary growth disturbance, and for incisors and canines to erupt [31]. The mean ABG operation timing in our study is similar to the conclusion of other studies, regardless of whether preoperative ortho treatment was performed (Table 5). Preoperative orthodontic treatment starts approximately 5 to 6 months before ABG to optimize the position of the dentoalveolar structure, thus, enabling patients to achieve better oral hygiene and to reduce the interference while performing cleft mucoperiosteal dissection [32].

In our study, tooth agenesis is diagnosed in patients older than 7 y because the crown bud of the second molar is approximately completed. However, late mineralization of particularly the mandibular second premolar is observed in the normal population at the age of 10 y [33], and a delay of up to 0.7 y is reported for the cleft patients in comparison to non-cleft patients [34]. This should also be taken into consideration. After scanning a series of panoramic radiographs, tooth agenesis is diagnosed, with no sign of crown calcification and no history of dental trauma or extraction. The overall prevalence of tooth agenesis in our study is 71.9% (97/135), which is slightly lower than that reported in another Taiwanese study (73.4%) reported by Wu et al. [35]. Our results show that the upper lateral incisors have the highest rate of missing data, which is the same as reported in another study by Bartzela et al. [36]. In our study, missing upper lateral incisors are commonly found in cleft sites, which is in accordance with the study by Jamilian et al. [37].

Tooth shape anomalies occur exclusively on the cleft side, with mostly upper lateral incisors being malformed or peg-shaped. Previous research shows that the prevalence of microdontia in the general population varies from 1.5 to 2.0% [38]. In our study, the highest percentage of microdontia is observed in the upper left lateral incisors (18.9%), followed by the upper right lateral incisors (11.4%) (Figure 2). Both percentages are higher than reported in the general population and the unilateral CLCP patients (1.9~4.2%) reported by Akcam et al. [33], but lower than the Jordanian (37%) patients observed by Al Jamal et al. [14]. The micro-formed teeth can be restored to mimic the normal size, or extracted and replaced by the neighboring teeth depending on the severity of dental crowding, inter-arch relationship, and facial profile.

The percentage of transposed maxillary canines and first premolars is 10.6% in our study, which is similar to that reported in another study [39]. The general prevalence of tooth transposition is 0.33% in the meta-analysis conducted by Papadopulous et al. [40]. Cassolato et al. [41] found maxillary canines and first premolar transposition on the cleft side. In contrast, we observe six patients (4.5%) with transposition contralateral to the cleft side and one patient (0.8%) diagnosed with unilateral cleft lip and palate with bilateral transposition. Although the cause of transposition is not fully understood, the speculation of a higher prevalence in cleft patients may be due to the interaction between genetic factors, underdevelopment of the maxilla, and severe crowding.

A limitation of this study was the restricted sample size due to the exclusion of poor image quality and incomplete data collection. Hence, sufficient data records for future studies are necessary to incorporate larger sample sizes, which enhances the accuracy of prevalence evaluation.

## 6. Conclusions

In this study, no differences are found in the sex of patients with CLCP who underwent ABG operation on the affected sides. The most frequently missing teeth and microdontia are the maxillary lateral incisors in the cleft area of these patients. The prevalence of upper canine and first premolar transposition is higher in patients with CLCP than that in the general group.

The distribution of dental anomaly patterns in our study differs from those in other countries. The result may support the idea that the patterns of dental anomalies in CLCP patients are region-specific.

## Figures and Tables

**Figure 1 jpm-12-01708-f001:**
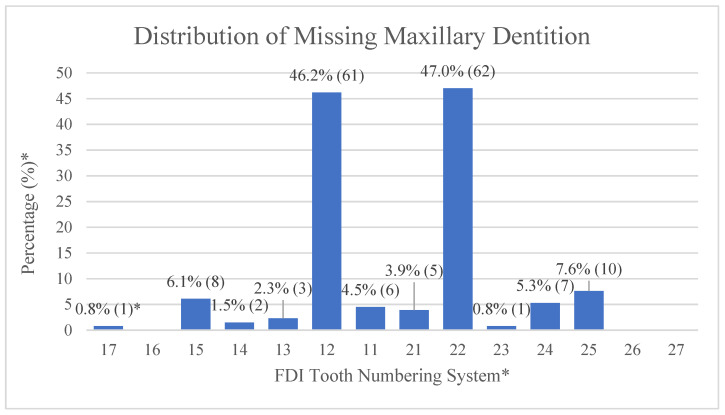
The distribution and frequency of missing teeth. * (1) Tooth numbering is according to the Fédération Dentaire Internationale (FDI) tooth numbering system for maxillary and mandibular dentition. (2) Percentage of missing teeth (missing number of each tooth site/total patient number (132)). (3) Numbers of missing teeth for each tooth site are listed in the parentheses. (4) The third molars are not examined in this study.

**Figure 2 jpm-12-01708-f002:**
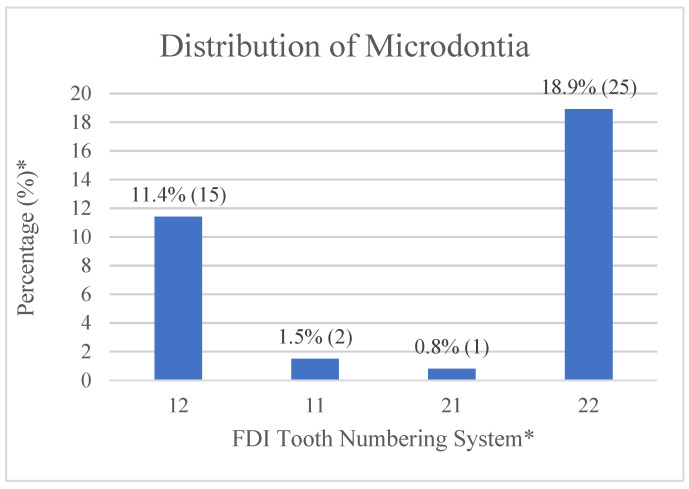
The distribution of microdontia. * (1) Tooth numbering is according to the Fédération Dentaire Internationale (FDI) tooth numbering system for maxillary and mandibular dentition. (2) Percentage of microdontia (number of each tooth site/total patient number (132)). (3) Numbers of microdontia for each tooth site are listed in the parentheses. (4) The third molars are not examined in this study.

**Table 1 jpm-12-01708-t001:** Gender distribution and mean age undergoing alveolar bone graft (ABG).

			ABG
	Numbers	Percentage (%)	Age (y) (Mean ± SD)
BOY (B)	82	62.1%	9.28 ± 0.80
GIRL (G)	50	37.8%	9.26 ± 0.68
TOTAL (B + G)	132	100%	9.27 ± 0.76

**Table 2 jpm-12-01708-t002:** Site distribution of cleft lip and palate. Patients with unilateral cleft lip and palate is subdivided into right and left side.

	Numbers	Percentage (%)
Unilateral (Uni)		
Right side	47	35.6
Left side	56	42.4
Total number	103	78.0
Bilateral (Bi)	29	22.0
Total number (Uni + Bi)	132	100

**Table 3 jpm-12-01708-t003:** Distribution of gender and relationship between gender and the number of affected sides.

	Boy	Girl
Unilateral (Uni)	55	37
Bilateral (Bi)	27	13
Total number (Uni + Bi)	82	50
*p* value	0.400	

**Table 4 jpm-12-01708-t004:** Distribution of gender and relationship between gender and the affected side.

Unilateral Cleft Lip and Cleft Palate		
	Boy	Girl
Right side	25	17
Left side	30	20
Total number	55	37
*p* value	0.962	

**Table 5 jpm-12-01708-t005:** Patient distribution in whether pre-ABG orthodontic treatment was received and the mean age of orthodontic treatment and ABG.

			ABG	Pre- ABG Ortho *
	Numbers	Percentage (%)	Age (y) (Mean ± SD)	Age (y) (Mean ± SD)
With pre-ABG ortho treatment	54	40.9	9.13 ± 0.70	8.72 ± 0.70
Without pre-ABG ortho treatment	78	59.1	9.37 ± 0.78	-

* Pre-ABG ortho treatment includes tooth sites of upper anterior teeth (tooth 11 to tooth 21 or tooth 12 to tooth 22) only. Tooth numbering is according to the Fédération Dentaire Internationale (FDI).

**Table 6 jpm-12-01708-t006:** Side distribution of transposition of upper canine and first premolar. (Patient with unilateral side of transposition is subdivided into ipsilateral or contralateral to the cleft site.).

	Numbers	Percentage * (%)
U3 and U4 transposition at the ipsilateral side	7	5.3
U3 and U4 transposition at the contralateral side	6	4.5
U3 and U4 transposition at both side	1	0.8
Total of U3 and U4 transposition	14	10.6

* Percentage of transposition of upper canine and first premolar (number of each tooth site/total patient number (132)).

## Data Availability

The data presented in this study are available upon request from the corresponding author. The data are not publicly available, due to the privacy protection of the studied individuals.

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
