# Peer review of "Prevalence of Dental Anomalies in Taiwanese Children with Cleft Lip and Cleft Palate"

_jpm, 2022, doi:10.3390/jpm12101708_

Round 1

Reviewer 1 Report

Although this study is interesting, there are some flaws in this study.

It has been written that just Hemi-facial macrosomia and Pierre Robin syndrome were excluded from the study. What about other syndromes? It means how many patients were syndromic or not syndromic.

I suggest you complement tables 4 and 5 by incorporating graphics. Graphics will be in a more visual/educational way

Chi-square tests must be used to assess the statistical significance between variables.

Were the results of the two observations blinded to each other?

What test was used for the Interobserver agreement? And what was the result of it?

To expand the discussion, the following article must be cited

Cleft sidedness and congenitally missing teeth in patients with cleft lip and palate patients Progress in orthodontics 17 (1), 1-4

Reviewer 2 Report

The manuscript with the title „Prevalence of dental anomalies in Taiwanese children with cleft lip and cleft palate” is based on an interesting topic, but needs major revisions.

I recommend taking into consideration several changes.

Please see my suggestions below.

Introduction

·      The Introduction is too short and does not contain sufficient information on the topic of dental anomalies. I suggest adding some references that define and explain the different dental anomalies, rather than just listing some anomalies.

·      The rationale for this study is missing. It is important for the reader to know why was this study important and why it was needed to be conducted.

·      The aim of this study should be developed. It is again too short and is lacking important information. Although, the authors affirm that: The aim of this study was to investigate the prevalence of tooth agenesis, malformation of teeth, the eruption pattern of upper canines/first premolars in Taiwanese children”, much of the Results section focuses on ABG. This should also be highlighted in the aim.

Materials and methods

·      This section should be divided into several subsections for a better understanding and in order to increase readability.

·      The study consisted of 132 patients with cleft lip and palate – This belongs to the Results and should be removed from this section.

·      The inclusion and exclusion criteria are insufficient.

·      The enrolled list contained 132 patients (boy: 82; girl: 50) who had received alveolar bone grafting (ABG) due to insufficient alveolar bone from cleft side. The mean ABG operation age of our patient was 9.27±0.76 y/o (range: 7.04 ~ 12.58 y/o)– This belongs to the Results and should be removed from this section.

·      The section requires some information regarding the person/persons responsible for the analysis of records, pictures, and x-rays.

·      The statistical analysis must be better performed. No statistical tests were performed (no correlations, no comparisons).

Results

·      The Results must contain data regarding the initial number of patients, the number of excluded patients, and finally, the final number of patients.

·      The section must contain information regarding the mean age of the patients included in this study.

·      The Results must be improved with correlations and comparisons, sustained by statistical tests, so that the reader could see whether or not the results obtained are statistically significant or not.

Conclusions

·      I don’t recommend using percentages in the Conclusions section.

·      This section should be shortened and resumed.

Best regards!

Round 2

Reviewer 2 Report

Thank you for addressing my comments!

I consider the manuscript to be suitable for publication.

Best regards!